# Increasing Levels of Supplemental LED Light Enhances the Rate Flower Development of Greenhouse-grown Cut Gerbera but does not Affect Flower Size and Quality

## David Llewellyn, Katherine Schiestel and Youbin Zheng *

School of Environmental Sciences, University of Guelph, 50 Stone Road East, Guelph, ON N1G 2W1, Canada; dllewell@uoguelph.ca (D.L.); katie.schiestel@gmail.com (K.S.)

* Correspondence: yzheng@uoguelph.ca

**Abstract:** To investigate the influence of supplemental lighting intensity on the production (i.e., rate of flower development, flower quality, and yield) of cut gerbera during Canada's supplemental lighting season (November to March), trials were carried out at a research greenhouse. Five supplemental light emitting diode (LED) light intensity (LI) treatments provided canopy-level photosynthetic photon flux densities (PPFD) ranging from 41 to 180 μmol m$^{-2}$ s$^{-1}$. With a 12-h photoperiod, the treatments provided 1.76 to 7.72 mol m$^{-2}$ d$^{-1}$ of supplemental light. Two cultivars of cut gerbera (*Gerbera jamesonii* H. Bolus ex Hook.f) were used to evaluate vegetative growth and flower production. Plugs of 'Ultima' were assessed for vegetative growth and rate of flower development. There were minor LI treatment effects on number of leaves and chlorophyll content index and flowers from plants under the highest versus the lowest LI matured 10% faster. Reproductively mature 'Panama' plants were assessed for flower yield and quality. 'Panama' flowers from the highest LI treatment had shorter stems than the three lowest LI treatments, and flowers from the middle LI treatment had larger diameter than the other treatments. Flowers from the lowest LI treatment had lower fresh mass than the three highest LI treatments. There were linear relationships between LI and numbers of flowers harvested, with the highest LI treatment producing 10.3 and 7.0 more total and marketable flowers per plant than the lowest LI treatment. In general, increasing levels of supplemental light had only minor effects on vegetative growth (young plants) and size and quality of harvested flowers (mature plants), but flowers from plants grown under higher LIs were more numerous and matured faster.

**Keywords:** flower bud development; flower number; flower quality; *Gerbera jamesonii*; growth; DLI

## 1. Introduction

In greenhouses at higher latitude regions, such as northern USA and Canada, it is often considered necessary for growers of year-round commodities (e.g., cut flowers) to use supplemental lighting to meet the crops' economic minimum lighting requirements during the darker months, due to low natural light conditions and short daylengths. While many economic (e.g., capital cost of fixtures and electricity prices) and practical (e.g., fixture positioning and capacity of electrical supply infrastructure) elements are considered when outfitting a greenhouse with supplemental lighting systems, the response of the crop(s) to additional lighting is a key factor which can only be evaluated through careful production trials.

The photosynthetic responses of plants to increasing levels of photosynthetically active radiation (PAR), generally described in terms of photosynthetic photon flux density (PPFD, μmol m$^{-2}$ s$^{-1}$), have been well established for many plant species and environments. When considering supplemental

lighting in greenhouse production scenarios, crops are generally subjected to light intensities (LI) that are on the linear portion of the photosynthetic light response curve (i.e., far lower than LIs needed to saturate the photosynthetic machinery). By extension, the yield responses of many greenhouse crops are commonly generalized as being directly proportional to the levels of light provided, with every 1% increase in lighting resulting in concomitant 0.5% to 1% increases in production [1]. This relationship has borne out for some economically relevant production indices in various floriculture commodities, such as cut gerbera [2,3], potted begonias [4], and cut roses [5,6].

For optimization of commercial greenhouse production scenarios which utilize supplemental lighting, it is necessary to determine the impact supplemental daily light integral (DLI), supplied within practical constraints of intensity and photoperiod, has on economically relevant production indices for target commodities. This may be especially relevant for cut flower production, such as cut gerbera, where the harvestable product represents a relatively minor component of total plant biomass production. Therefore, yield may be less directly related to rates of photosynthesis or carbon assimilation, while other crop x lighting interactions, such as photomorphogenic effects and flower bud development, may become more relevant with increasing levels of supplemental DLI [7]. We are unaware of any other references which have directly investigated the effect of DLI on vegetative growth, days between transplant and first visible flower bud, rate of flower development from visible bud to harvest, or fresh harvest metrics of cut gerbera flowers.

Auito [2] investigated cut gerbera production under a range of supplemental PPFD and photoperiods provided by high pressure sodium (HPS) lights. Their results showed that 12-h photoperiod maximized flower production for a given supplemental DLI. Conversely, Pettersen and Gislerød [8] found that a 20-h photoperiod had a higher production of cut gerbera than a 10-h photoperiod. However, the trials were done in a growth chamber, with a fixed PPFD, thus confounding the effects of photoperiod and DLI (i.e., the 20-h photoperiod had twice the DLI as the 10-h photoperiod), making it difficult to extrapolate their results to greenhouse environments. In a parallel study, Auito [2] also found linear or near linear relationships between supplemental light intensity and cut gerbera production in a greenhouse, using PPFD levels ranging from 75 to 300 $\mu$mol m$^{-2}$ s$^{-1}$ with a 12-h photoperiod (i.e., DLIs of 3.2 to 13.0 mol m$^{-2}$ d$^{-1}$). However, natural lighting was only reported as seasonal mean values for outdoor DLI throughout the 6-month trial period ($\approx$ 7.4 mol m$^{-2}$ d$^{-1}$, November to April) with an (estimated) greenhouse transmission value of 50%. Therefore, it is not possible to draw conclusions based on the absolute light levels (i.e., natural + supplemental) within the treatment plots. Approximate values for natural DLI at crop level in this study would probably have averaged between 3 and 4 mol m$^{-2}$ d$^{-1}$ (based on 50% of 7.4 mol m$^{-2}$ d$^{-1}$), which is similar to the winter lighting conditions in the research greenhouse facility used in the present study [9]. Spanomitsios et al. [3] found a positive linear relationship between mean daily solar radiation and rate of cut gerbera flower production. In this study, the slope of the relationship between light and production (slope = 0.47) indicated that $\approx$ 0.5% increase in flower production could be expected for every 1% increase in total light. However, it takes approximately four weeks for a cut gerbera flower to mature from visible bud to harvestable stage. Therefore, the reported relationship between daily net radiation and flower yield would have been more realistically portrayed if harvest data had been related to the average DLI for the four weeks prior to each harvest. Further, it is not possible to infer PPFD or DLI at crop level in this study as it is not clear how or where light data were collected or how the data were processed. Mustapić-Karlić et al. [10] found a positive influence of supplemental lighting on flower yield of two cut gerbera cultivars. They compared treatments of natural lighting with natural + supplemental HPS lighting providing $\approx$ 3 mol m$^{-2}$ d$^{-1}$ of additional PAR (i.e., PPFD of $\approx$ 70 $\mu$mol m$^{-2}$ s$^{-1}$ with a 12-h photoperiod). However, the DLI at crop level is unknown because the natural light levels at crop level were not reported. Similarly, Gagnon and Dansereau [11] found increases in potted gerbera productivity and reductions in time to flowering with increasing levels of supplemental HPS lighting (ranging from 1.7 to 5.2 mol m$^{-2}$ d$^{-1}$). However, the authors also did not report natural light levels, making it impossible to draw conclusions about the absolute influence of the lighting

treatments on production. While these trials clearly indicate positive relationships between increasing levels of supplemental lighting and production of cut gerbera, insufficient information on canopy-level lighting conditions make it difficult for readers to critically evaluate the total amount of PAR received by the plants in these trials [7].

With respect to the quality of supplemental light, research has shown that at similar PPFD, supplemental PAR from light-emitting diode (LED) technologies have resulted in similar crop production metrics as traditional HPS in greenhouse commodities, such as leafy vegetables [12], fruiting vegetables [13–15], ornamentals [16–18], and cut flowers [19]. While the capital costs of LED technologies are still considerably higher than HPS, LEDs have many advantages over HPS. LEDs can provide narrow wavebands of light specifically targeted at the maximum absorption bands of photosynthetic machinery. LEDs are touted to have greater than twice the lifetime as HPS and also have the potential to achieve higher efficacies (i.e., conversion of electricity into PAR). Moreover, LEDs are naturally dimmable, providing the capacity to adjust intensity according to natural lighting conditions, as well as on-demand customization of spectral recipes, providing greater plasticity for photoperiod and photomorphological control within a single fixture [20,21]. Accordingly, leading researchers and industry professionals consider it only a matter of time before LED technologies replace HPS as the benchmark technology for supplemental lighting in greenhouse applications [22].

The objectives of this study were to evaluate the relationships between increasing levels of supplemental lighting from LEDs during the darker months in Canada on the growth, flower development, yield, and quality of greenhouse grown cut gerbera.

## 2. Materials and Methods

### 2.1. Location, Trial Bench, and Greenhouse

The study took place at the University of Guelph in Guelph, ON, Canada, (43.55 ° N, 80.25 ° W) beginning on 9 November 2015 and ending on 25 February 2016 (i.e., 107 d). The study was set up within a single $7.2 \times 7.2$-m glass-clad research greenhouse compartment, containing four $4.57 \times 1.07$ m benches, with 0.91 m spacing between them. The long sides of the benches were positioned in an east-west direction (i.e., parallel with the track of the sun).

### 2.2. Lighting Treatments and Plant Distribution

There were five PPFD treatments, two pots of plants (i.e., two subsamples) for each of two cultivars under each PPFD treatment on each bench, as well as four replicates (i.e., benches) within the greenhouse compartment.

There were four LED fixtures (Pro 325; LumiGrow, Novato, CA, USA) per bench, located 30 and 100 cm (measured on-center of each fixture's LED array) from both ends of each bench. The lights were centered along the long axis of the bench and fixed with the LED arrays 140 cm above pot level. Each fixture was affixed with shrouds arranged parallel with the long sides of the benches made of white vinyl siding (Cedar Creek D4D; Abtco, Milton, ON, Canada) to reduce stray lighting from adjacent benches. The fixtures were set with an area-averaged photon flux ratio of blue (B, 400 to 500 nm) to red (R, 600 to 700 nm) of B22:R78. Fixture positioning and mapping light distribution patterns were done at night using a radiometrically-calibrated spectrometer (USB2000+; Ocean Optics, Dunedin, FL, USA) coupled to a 400-μm diameter UV-VIS optical fiber with a CC-3 cosine corrector (Ocean Optics, Dunedin, FL, USA). Light distribution (intensity and quality) was measured at pot level on a $2 \times 12$ rectangular grid (i.e., 24 specific locations), centered on the geometric center of the bench, with 30 cm separating adjacent measurement locations. For the trial, individual cut gerbera pots were centered on each of these bench locations and remained there for the duration of the trial. In this configuration, the supplemental light treatment at pot level of each plant was kept at a constant, known value. This design resulted in five unique supplemental PPFD treatment levels on each bench (labeled T1 to T5).

Two cut gerbera (*Gerbera jamesonii* H. Bolus ex Hook.f) cultivars, 'Panama' and 'Ultima', were used for this trial. 'Panama' plants were sourced from an active production environment ($\approx$ 5 months of active flower production) from a local grower (Bayview Flowers Ltd., Lincoln, ON, Canada). Flower stems longer than 2.5 cm were removed from 'Panama' plants at the beginning of the study. 'Ultima' plants came from the supplier, Florist Holland B.V. (De Kwakel, The Netherlands), as 'Jiffy 4' plugs.

On 8 October 2015, the plugs were transplanted into round 19 cm diameter × 19 cm tall pots; filled with coarse coir mix typically used by and obtained from a local cut gerbera grower. 'Ultima' plants began the trial in the vegetative stage, with no visible flower buds. Equal numbers of plants from each cultivar were positioned on the benches such that the cultivars were arranged in an alternating fashion. This arrangement resulted in two plants of each cultivar per treatment per bench, plus two border plants on the ends of each bench. The planting density was $\approx$ 7 plants m$^{-2}$, which was consistent with local commercial cut gerbera greenhouses. Although the location of each plant was fixed, the plants were rotated one-quarter turn weekly to reduce pot-location effects.

### 2.3. Environmental Management

The greenhouse environment parameters were set at similar levels to those used by local cut gerbera producers. Supplemental LED lighting was turned on daily 12 h before dusk and turned off at dusk, resulting in a constant 12-h photoperiod. Day and night temperature setpoints were 21 and 14 °C, respectively. Relative humidity (RH) was maintained at 70% using an aerial fogger system located at gutter level. Temperature and humidity dataloggers (HOBO U12-013; Onset Computer Corporation, Bourne, MA, USA) were located at canopy level in the center of each bench. PAR sensors (SQ-110; Apogee Instruments Inc., Logan, UT, USA) were located 1.75 m above the center of each bench (i.e., just above the top of the LED fixtures) and connected to the HOBO dataloggers. Temperature, RH, and PPFD were logged every 120 s throughout the study. Previous light uniformity data, collected by simultaneously logging the natural PPFD at fixture-level and bench-level (supplemental light fixtures present but left off) during a prior supplemental lighting season (i.e., November to March), indicated strong correlations in DLI measured between bench- and fixture-level locations on each bench. Coefficients relating natural DLI at fixture-level to bench-level derived from these data (not shown), were applied to the fixture-level PPFD data collected during the present trial to determine natural DLIs at canopy level on each bench.

### 2.4. Irrigation Management

Plants were drip irrigated using 20N-3.5P-16.6K All Purpose water soluble fertilizer (250 ppm N, pH 5.5; Plant Products Co. Ltd., Brampton, ON, Canada) with temporary substitutions of well water (pH and EC of 7.9 and 1000 $\mu$S cm$^{-1}$, respectively), when necessary, to maintain an approximate root zone pH of 5.5 and EC of 2500 $\mu$S cm$^{-1}$. Pulse irrigation occurred every second day, at 0915 and 1315 HR for 180 s each time. This irrigation protocol was aimed at producing approximately 10% to 25% leachate. Hand-watering was used as needed to supplement the drip irrigation.

### 2.5. Plant Growth, Leaf Chlorophyll Content Index, Flower Quality, and Yield Metrics

The number of leaves and chlorophyll content index (CCI) were measured approximately monthly on each 'Ultima' plant using a chlorophyll meter (CCM-200 Plus; Opti-Sciences, Hudson, NH, USA). CCI measurements were taken (three measurements per leaf with the average CCI value recorded), near the leaf margin (i.e., avoiding larger venation) of the youngest fully-expanded leaf of each plant. 'Ultima' plants were also checked twice weekly for the development of flower buds. Once each stem was ≥ 1 cm long, it was tagged with a unique identifier and the respective date was recorded as the date of appearance. This provided the days from transplant to first visible flower bud (i.e., stems ≥ 1 cm), as well as insight into the rate of flower development (i.e., the time between visible flower bud appearance and harvest).

Flowers on 'Panama' plants were harvested twice weekly. Flowers were deemed harvestable once they developed one complete ring of matured anthers. Fresh mass, flower diameter (measured petal tip to petal tip on the widest part of the flower), and stem length (measured from heel to the base of the flower) were measured on each harvested flower. Flower quality was also classified subjectively as either marketable or unmarketable according to the severity of malformations and pest damage.

*2.6. Statistical Analysis*

The experiment was a block design with 5 treatments and 4 concurrent replications. All data sets were analyzed using JMP® (version 13; SAS Institute Inc., Cary, NC, USA, 1989–2017). Least squares analysis was used for light treatment uniformity; vegetative growth, rate of appearance of visible flower buds, and flower development metrics in 'Ultima'; and accumulated total and marketable flowers harvested per plant in 'Panama'. Flower yield metrics in 'Panama' were analyzed using the Mixed-Models add-in, which accounts for the different numbers of flower stems harvested from each plant. Data were evaluated using a significance level of $p \leq 0.05$ using Tukey's honestly significant difference (HSD) test. Days between the appearance of flower buds and harvest on 'Ultima' and accumulated total and marketable flowers harvested per plant on 'Panama' underwent regression analysis ($p \leq 0.05$), using total DLI (i.e., natural + supplemental) as the independent variable.

### 3. Results

Weekly average canopy-level natural DLI for the 17-week trial ranged from $\approx$ 1 to 6 mol m$^{-2}$ d$^{-1}$ with an overall average of 3.6 mol m$^{-2}$ d$^{-1}$ (Figure 1), which was consistent with previous years' light characterizations within the same experimental greenhouse (data not shown). Daytime (i.e., daily timeframe when supplemental lighting was on) and nighttime (i.e., daily timeframe when supplemental lighting was off) temperatures were (mean ± SD) 20.4 ± 2.0 °C and 16.6 ± 1.24 °C, respectively.

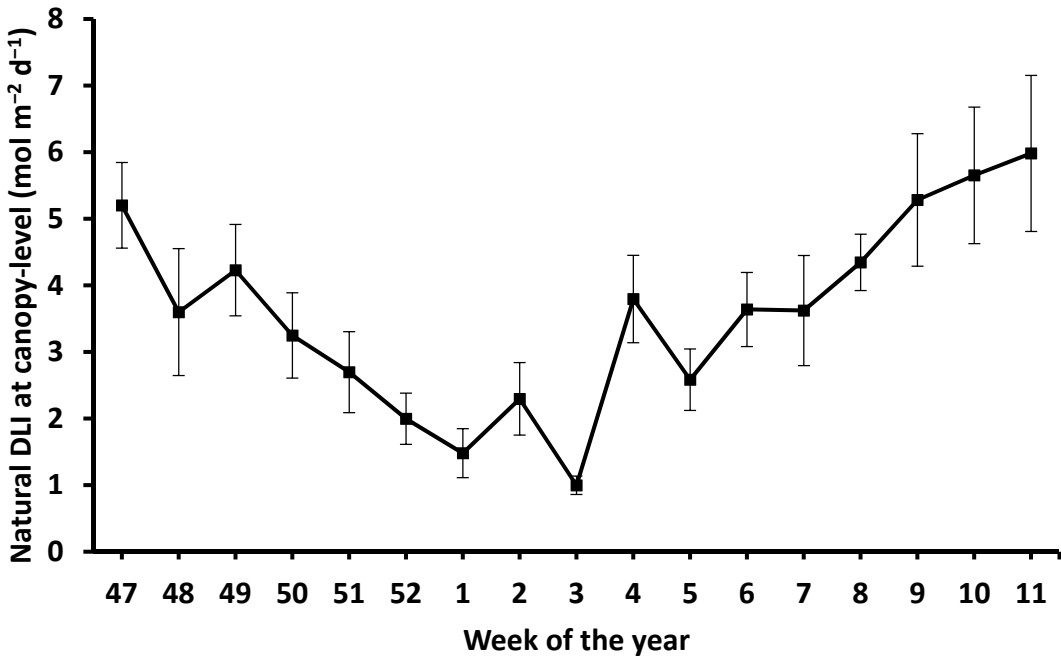

**Figure 1.** Weekly natural daily light integral (DLI) at canopy level (average ± SE, n = 7). The overall average natural DLI, during the 17-week trial, was 3.6 mol m$^{-2}$ d$^{-1}$.

The supplemental PPFD treatments ranged from 40.7 to 179 µmol m$^{-2}$ s$^{-1}$, corresponding to 1.8 to 7.7 mol m$^{-2}$ d$^{-1}$ of daily supplemental PAR with a 12-h photoperiod (Table 1).

**Table 1.** Canopy-level supplemental photosynthetic photon flux density (PPFD) of the five supplemental light-emitting diode (LED) treatments and their associated supplemental and total daily light integrals (DLI).

| Treatment | PPFD ($\mu$mol m$^{-2}$ s$^{-1}$) | | | DLI (mol m$^{-2}$ d$^{-1}$) | |
| :---: | :---: | :---: | :---: | :---: | :---: |
| | Mean | Max | Min | LED [y] | Total [x] |
| T1 | 40.7 ± 1.3 a [z] | 45.2 | 33.2 | 1.8 | 5.3 |
| T2 | 76.1 ± 1.6 b | 87.4 | 64.1 | 3.3 | 6.9 |
| T3 | 133 ± 2.4 c | 151 | 114 | 5.7 | 9.3 |
| T4 | 167 ± 1.9 d | 181 | 153 | 7.2 | 10.8 |
| T5 | 179 ± 1.8 e | 192 | 162 | 7.7 | 11.3 |

[z] There were no block or bench position effects on supplemental PPFD within each treatment, so data are pooled means for each treatment ± SE (n = 16). Values in the same column followed by the same letter are not different at $p < 0.05$, using Tukey's honestly significant difference (HSD). [y] DLI from supplemental LEDs were calculated using mean PPFD from each treatment and 12-h photoperiod. [x] Total DLI is the sum of supplemental DLI from LED treatments and experiment-wise mean natural DLI of 3.6 mol m$^{-2}$ d$^{-1}$.

'Ultima' plants chosen for each treatment had uniform CCI and number of leaves at the start of the trial (9 November 2015). After one month of treatment (8 December 2015), plants in T5 had ≈ 4 more leaves than plants in T4. After two months of growth under the supplemental light treatments (6 January 2016), plants in T4 had higher CCI values than T1, T2, and T3, and plants in T5 had ≈ 6 more leaves than plants in T2 (Table 2).

**Table 2.** Chlorophyll content index (CCI) of the youngest fully-expanded leaf, and number of leaves per plant, measured at ≈ 4-week intervals post-transplant of 'Ultima' plants.

| Date | Treatment [total DLI (mol m$^{-2}$ d$^{-1}$)] | CCI | No. of Leaves |
| :---: | :---: | :---: | :---: |
| 9 November 2015 | T1 (5.3) | 43 ± 1.0 a [z] | 9.6 ± 0.96 a |
| | T2 (6.9) | 38 ± 1.6 a | 9.5 ± 0.62 a |
| | T3 (9.3) | 40 ± 2.2 a | 9.3 ± 0.65 a |
| | T4 (10.8) | 38 ± 1.8 a | 7.8 ± 0.67 a |
| | T5 (11.3) | 40 ± 1.9 a | 8.1 ± 0.83 a |
| 8 December 2015 | T1 (5.3) | 47 ± 1.5 a | 11.8 ± 1.8 ab |
| | T2 (6.9) | 47 ± 1.1 a | 11.6 ± 1.2 ab |
| | T3 (9.3) | 49 ± 1.5 a | 12.9 ± 1.2 ab |
| | T4 (10.8) | 48 ± 1.6 a | 10.8 ± 1.2 a |
| | T5 (11.3) | 49 ± 1.6 a | 14.8 ± 1.3 b |
| 6 January 2016 | T1 (5.3) | 47 ± 1.3 a | 17.5 ± 2.2 ab |
| | T2 (6.9) | 49 ± 1.0 a | 16.9 ± 1.9 a |
| | T3 (9.3) | 49 ± 1.2 a | 21.5 ± 2.7 ab |
| | T4 (10.8) | 50 ± 1.0 b | 21.3 ± 1.8 ab |
| | T5 (11.3) | 49 ± 1.0 ab | 23.1 ± 2.0 b |

[z] There were no block effects within each treatment at each measurement date, so data are pooled averages for each treatment ± SE (n = 8). Values in the same column with the same measuring day followed by the same letter are not different at $p < 0.05$, using Tukey's HSD.

Flowers in T5 matured (i.e., time between appearance of flower buds and harvest) ≈ 3.6 d faster than plants in T1, which represents ≈ 10% reduction in flower development time (Table 3).

There were only minor treatment effects in fresh flower harvest metrics on 'Panama' flowers (Table 4). Flowers grown in T5 had marginally shorter stems than flowers grown in T1, T2, and T3. Flowers grown in T3 were marginally larger and flowers grown in T1 were smaller than the other treatments (with < 0.2 cm difference in diameter). Flowers grown in T3 also had higher fresh mass than flowers grown in T1 and T2.

**Table 3.** Days between appearance of flower buds (i.e., stems ≥ 1 cm) and harvest for all 'Ultima' flowers harvested during the trial, for different total daily light integral (DLI) treatments.

| Treatment (total DLI (mol m$^{-2}$ d$^{-1}$) | No. of Days Between Visual Appearance of Flower Bud and Harvest |
|---|---|
| T1 (5.3) | 37.6 ± 0.90 a [z] |
| T2 (6.9) | 35.8 ± 0.99 ab |
| T3 (9.3) | 35.0 ± 0.86 ab |
| T4 (10.8) | 34.5 ± 0.83 ab |
| T5 (11.3) | 34.0 ± 0.71 b |

[z] There were no block effects, so data are pooled averages for each treatment ± SE (n = 8). Values in the same column followed by the same letter are not different at $p < 0.05$, using Tukey's HSD.

**Table 4.** Stem length, flower diameter, and fresh mass of 'Panama' flowers harvested throughout the trial, for different total daily light integral (DLI) treatments.

| Treatment (total DLI (mol m$^{-2}$ d$^{-1}$)) | Stem Length (cm) | Flower Diameter (cm) | Fresh Mass (g) |
|---|---|---|---|
| T1 (5.3) | 46.5 ± 1.10 a [z] | 9.9 ± 0.07 a | 19.3 ± 0.57 a |
| T2 (6.9) | 46.6 ± 1.05 a | 10.1 ± 0.06 b | 20.7 ± 0.52 ab |
| T3 (9.3) | 46.7 ± 1.05 a | 10.3 ± 0.06 c | 22.9 ± 0.51 c |
| T4 (10.8) | 45.9 ± 1.05 ab | 10.1 ± 0.06 b | 21.2 ± 0.51 bc |
| T5 (11.3) | 44.4 ± 1.04 b | 10.1 ± 0.06 b | 21.1 ± 0.52 bc |

[z] There were no block effects, so data are pooled means for each treatment ± SE (n = 8). Values in the same column followed by the same letter are not different at $p < 0.05$, using Tukey's HSD.

Regressing 'Panama' flower harvest numbers against total DLI indicated that every 1% increase in DLI increased cumulative flower yield by ≈ 1.5% (Figure 2). The trend was similar in terms of marketable flowers, where a 1% increase in DLI resulted in a concomitant ≈ 1% increase in the number of marketable flowers produced per plant.

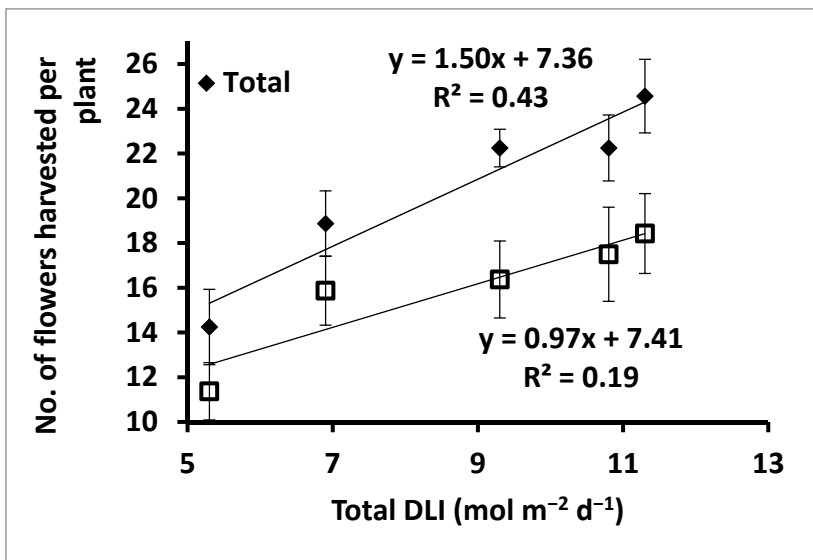

**Figure 2.** Cumulative total and marketable flowers harvested per plant, for 'Panama', in response to total daily light integral (DLI). Each point represents the treatment mean ± SE (n = 8); however, the equations are linear regressions of all of the harvest data on a per-plant basis.

## 4. Discussion

The range of supplemental PPFD levels used in this study raised the total canopy-level DLI to levels that approximately match the DLI range deemed necessary to produce minimum acceptable quality (6 mol m$^{-2}$ d$^{-1}$) to high quality (12 mol m$^{-2}$ d$^{-1}$) gerbera [23]. Vegetative growth and flower development indices were investigated using transplanted plugs of the 'Ultima' cultivar, while mature plants of the 'Panama' cultivar were used to assess the size, quality, and numbers of flowers produced.

There were no commercially-relevant LI treatment differences (or trends) in number of leaves or CCI of 'Ultima' plants. While there were also no LI treatment effects on the days from transplant to first visible flower (data not shown), flowers in T5 matured ≈ 10% faster than flowers in T1. Linear regression of the treatment means for days between appearance of first visible flower bud and harvest in 'Ultima' (in Table 3) against DLI indicates that each additional mol m$^{-2}$ d$^{-1}$ of DLI (e.g., ≈23 μmol m$^{-2}$ s$^{-1}$ of supplemental PAR over a 12-h photoperiod) shortened the time between flower bud appearance and harvest by 0.53 d. For example, adding ≈ 90 μmol m$^{-2}$ s$^{-1}$ of supplemental PAR with a 12-h photoperiod could shorten the flower production time by 2 d, during the darker months.

There were only minor (i.e., probably not commercially relevant) LI treatment effects on stem length, flower diameter, and fresh mass of marketable 'Panama' flowers. However, there were LI treatment effects on the total and marketable numbers of 'Panama' flowers harvested per plant, with plants in T5 producing ≈ 40% more flowers than plants in T1. Subjecting the cumulative flower production metrics to linear regression analysis showed that DLI could be used to predict the cumulative flowers produced per plant (Figure 2). Similarly, Bredmose [5,6] found linear relationships between supplemental light (HPS) intensity and numbers of flowers produced by mature plants of two rose cultivars, within the range of 0 to 174 μmol m$^{-2}$ s$^{-1}$. Auito's [2] investigation on the effects of supplemental light intensity and photoperiod on cut gerbera production is the most comprehensive to date. However, insufficient information was provided about the natural lighting environment under which the crops were grown; making it difficult to assess the actual lighting conditions (e.g., total DLI) in these trials. Despite this drawback, the author concluded that cut gerbera plants utilize supplemental light for flower production most efficiently at shorter photoperiods (i.e., 12 h), which is in line with local production practices. Auito [2] noted some cultivar-specific responses to increased supplemental PAR, although total flowers per plant and total dry mass generally increased linearly with increasing supplemental DLI (between 3.2 and 13.0 mol m$^{-2}$ d$^{-1}$, with a 12-h photoperiod)

In the present study, it was shown that doubling the total DLI from 6 to 12 mol m$^{-2}$ d$^{-1}$ by providing an additional 6 mol m$^{-2}$ d$^{-1}$ of supplemental PAR from LEDs could increase the number of flowers produced by nine flowers per plant (over 107 d). At typical commercial plant densities of 7 m$^{-2}$, this would result in monthly increases in flower production of ≈ 18 more flowers/m$^2$. In practical terms, if a grower provided 100 μmol m$^{-2}$ s$^{-1}$ of supplemental PAR, with a 12-h photoperiod, they could potentially increase the total number of flowers produced per plant during the darker months by ≈ 30%. To further contextualize in terms of energy cost, the efficacy factor of 1.29 μmol J$^{-1}$ for the LumiGrow Pro 325 fixtures used in this study [24] can be used to estimate that ≈ 1.3 kWh m$^{-2}$ d$^{-1}$ would be needed to add 6 mol m$^{-2}$ d$^{-1}$ of supplemental PAR from LEDs, which would be ≈ 2 kWh per additional flower produced, in the above scenario. However, the efficacy of some horticultural LED fixtures has more than doubled versus the fixtures used in this study [25], which would reduce the energy input per flower to less than 1 kWh for modern LED fixtures.

Future research should include broadening the range of commodities investigated under supplemental LED lighting intensity regimens, as well as investigating applications of targeted spectrum treatments (especially at night, where applicable) for manipulating crop morphology. A promising example of spectrum-mediated change in morphology are the increases in stem extension rates without some of the negative "shade avoidance" effects of high far red (700–800 nm) treatments by using low fluence rates of monochromatic blue light, applied at nighttime [26].

## 5. Conclusions

This investigation examined the influence of different levels of supplemental PAR, supplied by red and blue LEDs, on the production of cut gerbera during the darker months at higher latitudes. While there were few commercially-relevant LI treatment effects in the vegetative growth and harvested flower quality indices, higher light was shown to proportionally increase the rate of flower development and cumulative numbers of flowers produced. These relationships can be used by growers to assess the economic viability of using supplemental LED lighting to produce cut gerbera within their own production environments.

**Author Contributions:** Conceptualization, D.L. and Y.Z.; methodology, D.L., K.S. and Y.Z.; validation, D.L. and Y.Z.; formal analysis, D.L.; investigation, K.S. and D.L.; writing—original draft, D.L. and Y.Z.; writing—review & editing, D.L. and Y.Z.; visualization, D.L.; supervision, Y.Z.; funding acquisition, Y.Z. All authors have read and agreed to the published version of the manuscript.

**Funding:** This research was funded by the International Cut Flower Growers Association and the Joseph H. Hill Memorial Foundation, Inc.

**Acknowledgments:** LumiGrow, Inc. supplied the LED lighting fixtures. Plant materials were donated by Van Geest Bros Ltd (Ontario, Canada) and Bayview Flowers Ltd. (Ontario, Canada). None of these contributors were involved in the conduction of the study or submission of this article for publication.

**Conflicts of Interest:** The authors declare no conflict of interest. The funders had no role in the design of the study; in the collection, analyses, or interpretation of data; in the writing of the manuscript, or in the decision to publish the results.

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
