# Peer review of "Increasing Levels of Supplemental LED Light Enhances the Rate Flower Development of Greenhouse-grown Cut Gerbera but does not Affect Flower Size and Quality"

_agronomy, doi:10.3390/agronomy10091332_

Round 1

Reviewer 1 Report

General aspects: The manuscript fits within the scope of the journal. The manuscript is interesting. The title is clear and it is adequate to the content of the article. The results are presented clearly. The conclusions or summary are accurate and supported by the content.

I have some recommendation for authors:

  1. Present more clearly the purpose of the paper in the abstract and especially at the end of the introduction.
  2. From my point of view, the introduction is too large?
  3. Please see the citation style for this Journal. You must modify the whole text.
  4. The discussion chapter should be extended with comparative information. Currently, you have used only three sources in the discussions.
  5. Please highlight the degree of novelty and originality of the work.
  6. Include in the text potential research directions.

Reviewer 2 Report

Dear Authors, I find out the manuscript as of current interest, fitting with the scope of the journal. Introduction, Material and Methods paragraphs are well written and experiments are properly settled and described. Nevertheless I have to report some concerns on Results and Discussion Paragraph:

  • in the Results paragraphs you showed, in Table 3, timing of flowering for 'Ultima', but not the linear regression then discussed in Paragraph 4. As noted in comments directly written in the document, looking at Table 3, the data discussed on linear regression without showing the graph of, anyway the obtained formula and R2, does not appear as so clear (T2, T3 and T4 treatments are not significant different both to T5 and T1). You must improve Results paragraph and added these data. 
  • Figure 2 and Discussion: obtained formula and R2, especially for marketable flowers, can not properly support what was written in Discussion Paragraph. In my opinion Discussion can be considered a kind of speculation as collected data shown very high variability (see also reported SE) and the real increment in number of flowers starting by 7 DLI seems to be not so high (about 1.5% for each 1% of increased DLI) as reported in Result paragraph. By accurately reading the Graph of Fig. 2, especially for marketable flowers, it seems like that starting from T2 treatment (around DLI 7) number flower production reach a kind of plateau. Also flower quality does not appear particularly affected by different treatments. 

I suggest to improve results with data on linear regression about 'Ultima' flower timing (as reported in Discussion) and to carefully check the real effects of incremented DLI on flower production and quality, especially for marketable flowers which are those of greatest interest.

I have also some concerns about the selection of 'Ultima' for the assesement of the first steps of the crop (from transplanting to first harvest). 'Panama' was not available as young plants? Maybe you can improve Material and Methods about this selection (e.g., most widespread or appreciate cvs on the market?).

Finally, please, check carefully Instruction for Authors, especially for References that are not correctly reported in the manuscript.  

More details and comments are reported in the attached document. 

Kind regard

Reviewer 3 Report

This was a very easy to read article with a clear research objective, so thanks for that. Just some minor issues:

Line 54 "has" should be "has on"

Line 59 - evocation is a very strange word to use, I would go with "development" or some word more familiar to growers

Paragraph starting at Line 63 is much too long. Perhaps start new paragraphs at Line 79 and 88.

Line 148 should start a new paragraph

The R square for marketable flowers is quite low. Were either regressions significant? If not, perhaps leave out. 

Round 2

Reviewer 1 Report

The authors made the required changes. But the answer to my remarks is not sent point by point. The attached document refers only to the remarks of reviewer 1. However, analyzing the revised manuscript shows the changes suggested by me.

Reviewer 2 Report

Dear authors,

all notes and revisions aked by different reviewers were properly answered and corrected in the manuscript. 

Kind regard